# Assessment of the Development of Forest-Based Bioeconomy in European Regions

**Leire Barañano** [1], **Olatz Unamunzaga** [1,*], **Naroa Garbisu** [2], **Siebe Briers** [3], **Timokleia Orfanidou** [4,5], **Blasius Schmid** [6], **Inazio Martínez de Arano** [3], **Andrés Araujo** [7] **and Carlos Garbisu** [1]

1 NEIKER—Basque Institute of Agricultural Research and Development, Basque Research and Technology Alliance (BRTA), Parque Científico y Tecnológico de Bizkaia P812, 48160 Derio, Spain; lbaranano@neiker.eus (L.B.); cgarbisu@neiker.eus (C.G.)
2 Faculty of Economics and Business (Sarriko), University of the Basque Country (UPV/EHU), 48015 Bilbao, Spain; naroa.garbisu@gmail.com
3 European Forest Institute—Mediterranean Facility (EFIMED), St. Antoni Maria Claret 167, 08025 Barcelona, Spain; siebe.briers@efi.int (S.B.); inazio.martinez@efi.int (I.M.d.A.)
4 Bioeconomy Programme, European Forest Institute, Yliopistokatu 6B, FI-80100 Joensuu, Finland; cleo.orfanidou@efi.int
5 Department of Bioproducts and Biosystems, Aalto University, Vuorimiehentie 1, FI-02150 Espoo, Finland
6 Association of the Austrian Wood Industries, Schwarzenbergplatz 4, 1030 Wien, Austria; schmid@holzindustrie.at
7 Department of Management and Business Economics, University of the Basque Country (UPV/EHU), 48015 Bilbao, Spain; aaraujo@ehu.eus
* Correspondence: ounamunzaga@neiker.eus; Tel.: +34-944034300

**Abstract:** In recent years, the potential of the forest-based bioeconomy to provide competitiveness, differentiation, and sustainability to the European economy has often been claimed. Interestingly, regions, as territorial units with their own political and socioeconomic strategies, have been highlighted as the most suitable targets for the development of the European forest-based bioeconomy. Here, using the case method, we evaluated the development of the forest-based bioeconomy in three European regions (i.e., North Karelia in Finland, North Rhine-Westphalia in Germany, the Basque Country in Spain), by appraising the status of 10 previously identified key drivers through primary (interviews with experts) and secondary (literature review) sources of information. In our analysis, North Karelia and the Basque Country obtained the highest and lowest score, respectively, with regard to forest-based bioeconomy development. In any case, for the successful development of the forest-based bioeconomy in a European region, it is essential to accept the unnegotiable, critically, of the long-term sustainability of forest bioresources and production processes, as well as the need to foster the required changes in consumption patterns.

**Keywords:** bio-based economy; forestry sector; North Karelia; North Rhine-Westphalia; Basque Country

## 1. Introduction

Since the beginning of the Industrial Revolution, the discovery of fossil fuels has boosted unparalleled economic growth, with concomitant substantial beneficial effects for human well-being, quality of life, and social progress. Nonetheless, the use of fossil fuels for energy and materials has, also, resulted in considerable environmental degradation at a global scale, with climate change being one of the most negative consequences of such use. In this context, the bioeconomy has often been proposed as a constructive and suitable economic framework capable of reducing our strong and deep-rooted dependence on fossil fuels for energy and materials [1]. More precisely, the bioeconomy, defined as "the production of renewable biological resources and the conversion of these resources and waste streams into value added products, such as food, feed, bio-based products and bioenergy" [2], has repeatedly been claimed to be able to lead the way towards a more

sustainable economy and society. The bioeconomy notion was initially proposed and supported to effectively steer the desired transition from a fossil-fuel-dependent economy towards a more sustainable economy based on renewable biological resources [3]. In this context of thought, and in order to emphasize the criticality of sustainability and circularity for the long-term success of the bioeconomy field, a new definition of the bioeconomy was recently proposed: "a sustainable production and conversion of renewable biological resources and generated wastes into products and services, which fervently embraces ethics and circularity to simultaneously promote human well-being and nature conservation" [1].

In any event, desired transitions towards more sustainable attitudes, approaches, and methods are often characterized by a high degree of uncertainty, suspicion, and even scepticism among vested parties [4]. In consequence, pertaining to the often-proposed transition towards the bioeconomy, it is of paramount importance to precisely understand and come to terms with the predictably required far-reaching structural and functional changes (at various levels, e.g., policy, industry, market, consumption) in our socioeconomic systems, if we are to successfully address that shift [5].

From a theoretical perspective, in many aspects, the bioeconomy transition alludes to a sustainability transition, a term which refers to a "long-term, multi-dimensional and fundamental transformation process through which established socio-technical systems shift to more sustainable modes of production and consumption" [6]. Different sustainability transition frameworks have been proposed and used, such as, for instance, the multi-level perspective on socio-technical transitions, the strategic niche management, the transition management, the technological innovation systems, the techno-economic paradigm shifts, and the socio-metabolic transitions [7–9]. Sustainability transitions have several characteristics [10]: (i) they are goal-oriented; in this respect, since we are dealing with a collective good (i.e., sustainability), public authorities and civil society are crucial to address public goods and internalize negative externalities, to modify economic frame conditions, and to encourage green niches; (ii) more sustainable solutions often do not offer evident user benefits (after all, we are dealing with a collective good) and, besides, usually score lower on price/performance dimensions than already established technologies, thus requiring changes in economic frame conditions (e.g., taxes, subsidies, regulations, policies); and (iii) the empirical domains where sustainability transitions are most needed are characterized by large firms that possess complementary assets (e.g., specialized manufacturing capability, experience with large scale, access to distribution channels, service networks, complementary technologies, etc.), providing them with strong positions vis-à-vis innovative entrepreneurs.

In particular, the multi-level perspective (MLP) on socio-technical transitions posits that transitions occur through interactions within and between the following three analytical levels: (i) *niches*, or spaces where innovative activity takes place and where protection is offered from prevailing rules; (ii) *regimes*, or the socio-technical systems which include the network of actors and social groups, the binding rules, and the required technical-material elements; and (iii) a *socio-technical landscape*, which includes exogenous events and trends (e.g., demographic vicissitudes, macro-economic tendencies, political developments, wars and crises, cultural and societal values, climate change, etc.) [8]. Since transitions are often defined as shifts from one regime to another, the regime analytical level is of primary interest; thus, the niche and landscape levels are seen as derived concepts, since they are both defined in relation to the regime [10].

The main allure of the MLP (for instance, for the abovementioned bioeconomy transition) is that (i) it provides a rather straightforward way of ordering and simplifying the analysis of complex, large-scale transformations in production and consumption modes demanded by the sustainability paradigm; and (ii) it links innovation activities configured in niches with structural transformations in regimes [11]. Relevantly, it has been reported that studies of the sustainability transition towards the bioeconomy have mainly focused on wood-based and agriculture-based value chains [9]. In any event, since the transition to the bioeconomy is often associated with the co-evolution of economic, social, technological,

institutional, and ecological developments [12], the MLP to socio-technical change, which has frequently been used in similar transition studies as an archetype to iterate dynamics of socio-technical changes and long-term complex sustainability transitions [6,11,13], is a most suitable framework to lead the transition to a bioeconomy based on renewable biological resources. At a macro-level analysis, some aspects that must be taken into consideration for the bioeconomy transition are the increasing uncertainties, due to scarce resources and the current climate crisis, and escalating demands due to growing global human population, as well as changes derived from technology and scientific discoveries [14].

European regions, as territorial units that frequently display their own political and socioeconomic strategies, have often been highlighted as the most suitable targets for the implementation of the (forest-based) bioeconomy in Europe [15,16]. In this way, the (forest-based) bioeconomy has been linked with the notion of bioregionalism [1], which claims that socioeconomic and political systems are more sustainable when organized according to naturally defined areas called bioregions [17,18]. In particular, according to the European Commission [19], the transition to a bioeconomy-based model takes place gradually through the generation of initiatives at local and regional level.

Relevantly, the BERST Tool (https://berst.databank.nl/, accessed on 6 December 2021), designed by the EU Horizon 2020 Project "Building Regional Bioeconomies" [20], identified the following four pillars of bioeconomy readiness for a given region: biomass availability and land use; demographics and the quality of the workforce; employment and the structure of firms; and innovation. Positively, when evaluating the potential of a given region for (forest-based) bioeconomy development, its capacity to constantly and sustainably generate the required biological raw materials (biomass) at an appropriate rate and in the long-term, together with its innovation and entrepreneurship experience, must be seriously taken into account. Other pivotal factors at the regional level are the existing and upcoming legal frameworks, economic infrastructures, social demands, and, even, the culture and history of using natural renewable bioresources in that specific region [15]. Nonetheless, given the non-trivial uncertainties still associated to the identification and required magnitude and the degree of maturity of the main key drivers needed for the successful implementation and development of (forest-based) bioeconomy in a given region, far more research is needed, especially empirical research based on real case studies.

Since forest biomass is one of the most available bioresources to support bioeconomy growth in many European regions [21], in this study, using the case method, we evaluated the development of the forest-based bioeconomy in three European regions (i.e., North Karelia in Finland, North Rhine-Westphalia in Germany, the Basque Country in Spain) by appraising the status of 10 previously identified key drivers, grouped in four categories—institutional, demand, supply, and biomass-related drivers—through primary (interviews with experts) and secondary (literature review) sources of information. Our initial proposition was that, for the forest-based bioeconomy to successfully and sustainably develop in the long-term in a European region, many different drivers must be present and/or stimulated in unison by a variety of stakeholders (regional governments, research institutions, companies, consumers, etc.). It is, therefore, imperative to have an in-depth understanding of the identity and magnitude of these key drivers, in order to be able to establish, with a reasonable degree of certainty and exactness, the level of preparedness and development of the forest-based bioeconomy in a European region.

## 2. Materials and Methods

As indicated above, the main objective of this study was to assess the development of the forest-based bioeconomy in three European regions (North Karelia in Finland, North Rhine-Westphalia in Germany, the Basque Country in Spain). These three regions were selected for the following reasons: (i) they all belong to the Bioregions Facility of the European Forest Institute—EFI, which promotes transregional cooperation for a sustainable and integrative forest-based bioeconomy (this fact greatly facilitated access to the required information and contacts for the interviews); (ii) they all have a noteworthy

forestry sector, in terms of both land area covered by forests and level of support from regional administrations; (iii) they are all subjected to the politics, plans, and regulations of the European Union; (iv) they differ in terms of the percentage of the region surface covered by forests: North Karelia = 70%, North Rhine-Westphalia = 27%, and the Basque Country = 54%; (v) latitudinally speaking, they are located in different parts of Europe: North Karelia in the north, North Rhine-Westphalia in the centre, and the Basque Country in the south; (vi) they have different sizes (area in km$^2$): North Karelia = 21,584, North Rhine-Westphalia = 34,084, and the Basque Country = 7234; and (vii) they have different population sizes and densities: North Karelia = approximately 160,000 inhabitants and a density of 7.5/km$^2$, North Rhine-Westphalia = 17.9 million inhabitants and a density of 530/km$^2$, and the Basque Country = 2.2 million inhabitants and a density of 307/km$^2$. Regarding the availability of forest biomass, an unquestionable requirement for the development of the forest-based bioeconomy in a region, North Karelia has ca. 1.5 million hectares of forests and an estimated wood stock of 195 million m$^3$, with an annual growth of 8.9 million m$^3$ year$^{-1}$. North Rhine-Westphalia has a forest cover of 935,000 ha, an estimated wood stock of 277 million m$^3$, and an annual increase of 8.3 million m$^3$ year$^{-1}$. Finally, the Basque Country has 722,938 ha of forests, a wood stock of 62.6 million m$^3$, and an estimated annual growth of 3.4 million m$^3$ year$^{-1}$.

To this purpose, we initially carried out an analytical literature review in search for those key factors that can drive (forest-based) bioeconomy development at a regional level. The database "Science Direct" was selected as scientific source of bibliographic information, using the following anchor keywords: "forest bioeconomy" or "forest-based bioeconomy", "Europe" or "European region", "Drivers", and "Barriers". Our search incorporated publications for all years, but, given the relatively recent development of the field under study, a meaningful number of documents was only present from 2013 onwards. This databank search, conducted in November 2020, retrieved 154 publications, which were further screened to: (i) exclude book chapters (30 documents) and encyclopaedia (3 documents); and (ii) exclude documents not relevant for the main objective of this study: for example, publications where the concept of bioeconomy was based on energy production or publications that did not consider forest biomass as the main source of renewable biological resources. These selection steps resulted in a dataset of 39 publications [5,21–58], which were the basis for the identification of the 10 key drivers needed for the successful implementation and development of the forest-based bioeconomy in a given European region (see below). The list of publications is provided in Table A1. To this compilation of articles, we added a few reports [15,18,19], issued by relevant bodies, of key relevance for the topic under investigation.

Furthermore, to facilitate their use, analysis, and interpretation, the identified key drivers (see below) for forest-based bioeconomy development in a European region were grouped in four categories, based on [59]: (i) *institutional drivers*, dealing with strategic issues that are fundamentally developed by public authorities; (ii) *supply drivers*, related to the capacities and competences to generate products and solutions from and for the market; (iii) *demand drivers*, linked to market and consumer demands, which may or may not be susceptible to change as a result of awareness-raising campaigns and actions; and (iv) *biomass* related drivers, associated with the existence of a constant and sustainable supply of the required biomass, in terms of quantity, rate of generation and quality. Finally, we added two key drivers due to their current significance and practical relevance for European bioeconomy growth, expansion, and evolution: green public procurement and regional networks (see below). In this way, we ended up with 10 key drivers, grouped in four categories, for forest-based bioeconomy development in a European region: eight drivers identified during the literature review *plus* two drivers proposed by us according to our own analysis of the field.

After identifying and categorizing the key drivers, we proceeded with the assessment of forest-based bioeconomy development in the three abovementioned European regions by appraising the status of these 10 identified key drivers, through primary (interviews

with experts) and secondary (literature review; Table A2) sources of information, in the specific region under study.

Pertaining to the primary sources of information, between February and April 2021, a total of 32 interviews were conducted in Spanish (the Basque Country), German (North Rhine-Westphalia), and Finnish (North Karelia), face-to-face or by video calls (Microsoft Teams software). All the interviews were recorded, transcribed, and, then, translated into English. Regarding these interviews with experts (primary sources of information), apart from providing valuable qualitative information and knowledge on the topic, they were asked to score the 10 key drivers according to a Likert scale from 1 to 5 (i.e., considering the drivers as Likert items and scoring them from 1 to 5 according to their level of presence and degree of implementation in that specific region, with 1 and 5 corresponding to the lowest and highest extent of development for that specific driver, respectively). For the elaboration of these interviews, we followed the criteria and recommendations indicated by [60–63]. Regarding the selection of the specific persons to interview, we opted for (i) those who possess relevant information; (ii) among the informed, those who are most accessible; (iii) among the informed and accessible, those who are most willing to inform; and (iv) among the informed, accessible and willing, those who are most able to communicate the information accurately. Although it is true that, regardless of the type of research, a degree of subjectivity always exists, we avoided the use of pre-set answers in order to minimize the subjectivity often associated to the design of questionnaires [61]. A well-known pitfall associated with interviews is that they focus on and reflect the views of individuals (i.e., the interviewers and the interviewees) [63], pointing out to the need to be very cautious when interpreting the data and, above all, when extrapolating the conclusions. In any event, research, such as performed here through interviews with experts, involves the methodical and orderly collection, organisation, and interpretation of the obtained material [62], and must use strategies for (i) questioning findings and interpretations, (ii) assessing their internal and external validity, (iii) thinking about the effect of context and bias, and (iv) displaying and discussing the processes of analysis [62]. Besides, in the case of qualitative information, one must always take into consideration a variety of aspects and criteria that can affect the outcome of the analysis: (i) reflexivity ("the knower's mirror"); (ii) preconceptions ("the researcher's backpack"); (iii) theoretical frame of reference ("the analyst's reading glasses"); (iv) metapositions ("the participating observer's sidetrack"); and (v) transferability ("external validity") [62].

All the participants in the interviews: (i) are highly qualified professionals, e.g., general managers, innovation managers, or technology transfer managers; (ii) are professionally linked to the forestry value chain or to other value chains also related to the forest-based bioeconomy; and (iii) have a medium or advanced knowledge on the field of the forest-based bioeconomy. Importantly, we interviewed people from different backgrounds, in accordance with an approximately balanced ratio among the three following categories: (i) *researchers* working in universities, research institutes, or technology centres, with extensive experience in forestry science, biotechnology, and/or bioeconomy; (ii) *decision makers*, i.e., people working in public governmental institutions who define regulations and strategies related to environmental issues and/or agriculture-forestry management; and (iii) *business people* working in profit-oriented private companies linked to the forest value chain, including associations of forest owners, as well as packaging, construction, and renewable energy companies, whose main activity is based on the use of forest biomass. Although we did interview persons working in forest owner associations, we did not interview forest owners themselves. For future studies, it would be desirable to also include forest owners, as essential actors whose perspectives are critical for the successful development of regional strategies on the forest-based bioeconomy. Table 1 summarises the interviews with experts carried out in each European region by category.

**Table 1.** Interviews with experts by region and interviewee category.

| Category | North Karelia | North Rhine-Westphalia | Basque Country | TOTAL |
|---|---|---|---|---|
| Researchers | 3 | 3 | 5 | **11** |
| Decision makers | 2 | 1 | 2 | **5** |
| Business people | 5 | 5 | 6 | **16** |
| **TOTAL** | **10** | **9** | **13** | **32** |

Furthermore, according to data extracted from secondary sources of information (reports, documents, webpages, etc.), we also evaluated the status of the key drivers in the three regions, scoring them from 1 to 3 points, with 1 and 3 corresponding to the lowest and highest extent of development for that specific driver, respectively. Despite decades of research, the debate on the optimal number of response categories in rating scales is still unresolved [64]. As far back as 1970, Green and Rao [65] already described two groups regarding this debate on the optimal number of response categories in rating scales: one faction advocates for using fine grained scale points, whereas another faction, based on views about the respondents' capacity to discriminate between different points, advocates for only two or three response options; see, for instance, [66]. Though such debate is not within the scope of this study, here, we opted for a Likert scale from 1 to 5 for the primary sources of information versus a Likert scale from 1 to 3 for the secondary sources of information, since experts (in this case, experts on the specific situation of their corresponding region) can provide a variety of nuances and level of resolution, based on their knowledge of the subject matter, superior to that obtained through a literature review. In addition, oral and visual communication (interviews with experts were conducted via face-to-face or video calls) also provides much more nuance and degree of distinction, compared to reading bibliographical references.

After evaluating the status of each key driver according to the data provided by the primary and secondary sources of information, we reached a final conclusion on the extent of development (high: mean value $\geq$ 6 points; medium: 6 > mean value $\geq$ 4 points; and low: 4 > mean value $\geq$ 2 points) of the forest-based bioeconomy in the European region, where mean value was calculated as follows:

$$\text{Mean value} = \text{mean score extracted from the primary sources} + \text{mean score extracted from the secondary sources} \quad (1)$$

Since the scores given by experts ranged from 1 to 5, and the scores assigned by us ranged from 1 to 3, the highest and lowest possible mean values obtained by a given European region were 8 and 2, respectively.

## 3. Results and Discussion

### 3.1. Drivers for Forest-Based Bioeconomy Development in a European Region

As aforementioned, initially, we performed a literature review to identify the main key drivers for the development of forest-based bioeconomy in a European region. The most relevant studies are briefly discussed below.

Kardung et al. (2021) [59] proposed the existence of a series of primary forces that can condition the development of bioeconomy and which can be grouped in four categories: (1) *supply drivers:* technology and innovation (advances in biological sciences, advances in information and communication technologies, other technological advances), and market organization (advances in horizontal and vertical integration, globalization, increase in importance of climate change, anthropic pressure on ecosystems); (2) *demand drivers*: demographics, economic development, and consumer preferences; (3) *resource availability*; and (4) *measures of governments to influence bioeconomy development*: policies, strategies, and legislation (global, EU, and national policies; regional policies; legislation).

D'Amato et al. (2020) [22] provided valuable insights into the opportunities and challenges posed by the transition towards the bioeconomy: (1) in spite of its growing

popularity at policy and industry level, the concept of the bioeconomy is still weakly recognized; (2) the profitability of bioeconomy-based businesses is still low (they strongly rely on public support for research and development); (3) the assets for value creation, capture, and delivery include the renewable and circular nature of the biological resources, the compatibility of technological innovations with existing production and processing facilities, and the enabling potential of crucial partnerships with suppliers, producers, customers, and the whole innovation ecosystem; (4) a variety of social and environmental benefits for stakeholders external to the companies have been identified, such as, for example, job creation, improved quality of life, consumption choices of users and customers, and a lower social and environmental impact in production and the whole life cycle of the product/service; and (5) diversification of business models under the circular bioeconomy framework can help support sustainability goals. D'Amato et al. (2020) [22] agreed with Hansen (2016) [67] in defining small and medium-sized enterprises (SMEs) as key actors in the transition towards the circular bioeconomy, since SMEs' flexibility, dynamism, and ability to generate the necessary innovations can make the transition easier, as compared to larger companies in which such skills are often lacking. In addition, according to other studies [48,68,69], the development of new value chains with stakeholders, as well as the co-operation with international partners, actors with complementary knowledge, and the research community is crucial to successfully face the abovementioned transition. From an analysis of the works of [22,46,70,71], we extracted the following drivers: (1) a shared understanding and acceptance of bioeconomy; (2) high cooperation/transfer capacities between companies and industries; (3) a cohesive policy environment; (4) a mature market; (5) optimisation in the use of secondary flows; and (6) integration of industry with bioeconomy and sustainability strategies. Relevantly, Branscomb and Auerswald (2002) [72] pointed out the lack of financial resources for bioproduct innovation and production as a key barrier for bioeconomy implementation and development. In this respect, resources are generally more easily found for the research and development phase compared to the commercialization phase. The EU Horizon 2020 project entitled "Building Regional Bioeconomies" identified a set of criteria for the preparation and development of bioeconomy in a European region: (1) *essential criteria*: biomass availability, infrastructure, and governance; (2) *key criteria*: land use, cluster management and governance, commercialisation of innovative technologies, diffusion of technology, public support and acceptance, rate of SME formation, economic history, collaboration, entrepreneurial culture, and regulation; and (3) *desirable criteria*: domestic production of biomass, cluster size, R&D focused on key enabling technologies, consumer preferences, household income, availability of funding, proximity to financial institutions, presence of multinationals, quality of workforce, prominent universities or research institutes, intellectual property rights, trade policy, and size population. This EU H2020 project stressed that, in order to properly assess the potential of a European region for bioeconomy development, it is essential to pay attention not only to the quantity, quality, and rate of generation of the required biological resources (biomass, organic waste) in that specific region but, above all, to its capacities in terms of innovation and entrepreneurship.

Lier et al. (2019) [15] identified a number of key elements for bioeconomy development in a given region: (1) country- and region-specific socioeconomic and ecological settings; (2) legal framework; (3) social demand; and (4) a long history of using natural resources.

The European Commission (2017) [19] listed key factors for bioeconomy development at regional level: (1) abundance of natural and biological resources that can contribute to the generation of income and added value; (2) strong primary economic sectors (agriculture, fisheries, forestry); (3) important agro-food, fish, and wood/paper value chains with a strong technological specialisation; (4) relevant chemical or other industrial sectors that aim to shift from fossil resources to biological resources and bio-based products; (5) well-developed business sectors that seek for cooperation and public support in order to generate innovative new products; and (6) specialised higher education, research, and development centres that stimulate innovation for bioeconomy.

Näyhä (2020) [73] also identified a number of factors for the transition towards forest-based bioeconomy by a given region: (1) *key macro-environment factors of transforming forest-based companies*: global sustainability goals and challenges, aims for sustainable bioeconomy, supportive and "meeting the needs" of EU- and national-level policies and regulation, societal support for new businesses and entrepreneurship, experience and resources in the field, and value chain distribution; (2) *key industry and market environment factors of transforming forest-based companies*: high quality education and well-targeted training programs that meet the practical needs of the interested firms, interaction with customers and other stakeholders to truly understand their needs, partnerships, and spearhead products and companies; (3) *tangible organizational resources and capabilities needed for the transition of forest-based companies*: raw material issues (particularly resource efficiency), technical solutions, ability to scale up innovations, modern infrastructure, logistics, right location, and risk financing; and (4) *intangible and human-related resources and capabilities needed for the transition of forest-based companies*: innovative, agile, and encouraging organizational culture [56,73], knowledge of new markets, organizational cultures, communication, and marketing, flexible employees, multidisciplinary teams, non-hierarchical top management, and "power people" as a source of an innovative atmosphere [74]. In this line of thought, Evans and Salaiz (2019) [75] and Schoemaker et al. (2018) [76] emphasized the importance of having pro-active, flexible employees that can sense and seize new opportunities in fast changing business environments [77].

From this literature review, we extracted the following eight drivers for forest-based bioeconomy development in a given region (see description in Table 2): government plans and policies; research, development and innovation; training and talent; ecosystem for entrepreneurship; entrepreneurial capacities; existence of clusters; market awareness and demand; and biomass. From our own analysis, we included two additional drivers: green public procurement and the existence of regional networks. Pertaining to green public procurement, it is important to emphasize that public expenditure on works, goods and services represents approximately 19% of the EU Gross Domestic Product [78]. Public procurement is, therefore, a powerful tool to guide the market (e.g., the market for bio-based products) [79,80]. Likewise, the presence of regional networks is strongly linked to the European regional policy [81], which encourages cooperation between regions due to the many benefits associated to such collaboration, e.g., it helps ensure that borders are not barriers; it helps solve common problems; it facilitates the pooling and interchange of ideas and resources; and it enhances teamwork to achieve common goals.

**Table 2.** Drivers of the forest-based bioeconomy development in a European region.

| Driver | Description |
|---|---|
| *Institutional* | |
| **Government plans and policies** | The regional government has developed plans and policies on the bioeconomy and, in particular, the forest-based bioeconomy. These plans and policies are stable over time to guarantee the long-term sustainable development of the forest-based bioeconomy. |
| **Research, development and innovation** | The region has a solid, deep-rooted ecosystem for research, development, and innovation (R&D&i). Universities and technology centres develop R&D&I on the (forest-based) bioeconomy. There is a strong commitment and substantial investment in the (forest-based) bioeconomy. This fact is reflected in the regional RIS3 strategy. |
| **Training and talent** | There are specialised training programmes on the subject at all levels of education (schools, professional training, universities) in the region. There are regional programmes to attract talent intended to boost the development of the (forest-based) bioeconomy in its territory. |
| **Ecosystem for entrepreneurship** | There is a strong ecosystem for entrepreneurship with multiple factors and agents that interact to promote the creation of new businesses. The regional ecosystem for entrepreneurship stimulates the generation of new ideas, goods, services, and businesses. There are financing resources that support the ecosystem for entrepreneurship. |

**Table 2.** *Cont.*

| Driver | Description |
|---|---|
| *Institutional* | |
| **Green public procurement** | The regional government and public institutions promote green public procurement to encourage the development and implementation of sustainable products and services, such as those generated from the (forest-based) bioeconomy. |
| **Regional networks** | The region participates in European and international regional networks. In this way, the region is politically, commercially, and strategically connected to other regions with common interests and similar casuistry. These regional networks encourage cooperation between regions, e.g., regarding the development of the (forest-based) bioeconomy. |
| *Supply* | |
| **Entrepreneurial capacities** | Companies linked to the various value chains related to the (forest-based) bioeconomy exist in the region and have successful ad hoc business models. The region has the entrepreneurial capacity to evaluate the economic potential talent in a given item of new knowledge and to design ways to transform such potential into realizable economic value. The region displays individual and organizational capabilities that efficiently explore, integrate, and exploit untapped business opportunities. |
| **Existence of clusters** | Cluster or cluster-like initiatives related to the promotion and development of the (forest-based) bioeconomy are present in the region. These clusters are supported by a network of companies and institutions located in the region. The clusters are based on the region's unique assets for the (forest-based) bioeconomy. These clusters can encompass an array of industries and other entities such as suppliers of specialized inputs, providers of infrastructure, manufacturers of complementary products, trade associations, governmental and other institutions that can provide specialized training (vocational training), education (universities), legal, and technical support (agencies), etc. |
| *Demand* | |
| **Market awareness and demand** | The local–regional market and its consumers are actively demanding sustainable bio-based products. Ideally, these bio-based products should have the same or even better performance than those produced from fossil fuel-based raw materials. The society understands the concept of the (forest-based) bioeconomy and supports its implementation in the region, accepting the concomitant changes and consequences. If that is the case, many customers are willing to pay the extra cost of bio-based products provided they offset that economic disadvantage with other significant benefits: lower environmental impact, support of local businesses, and rural development. |
| *Biomass-related* | |
| **Biomass** | There is a sufficient and constant supply of (forest) biomass in the region in terms of quantity, quality, and rate of generation. The biomass is used in a sustainable way, encouraging ecosystem protection and biodiversity conservation. |

As indicated above, based on [22], these 10 drivers were classified in four categories (Table 2): institutional drivers, supply drivers, demand drivers, and biomass, related drivers.

### 3.2. Development of the Forest-Based Bioeconomy in Three European Regions

We evaluated the development of the forest-based bioeconomy in three European regions by assessing the status of the 10 key drivers, through primary (interviews with experts) and secondary (literature review) sources of information.

### 3.2.1. Primary Sources of Information: Interviews

For the interviews, the 10 key drivers were deployed in 15 items (see Table 3). Table 3 shows the scores given by the experts from the three European regions (North Karelia, North Rhine-Westphalia, the Basque Country) to the key drivers of the forest-based bioeconomy development, as well as the mean score value obtained by each region.

Table 3. Scores (from 1 to 5) for the different drivers given by the experts.

| Driver | Item | North Karelia | North Rhine-Westphalia | Basque Country |
|---|---|---|---|---|
| | *Institutional* | | | |
| **Government plans and policies** | Existence of stable policies and plans on the (forest-based) bioeconomy in the region | 4.4 | 3.5 | 3.6 |
| | Allocation of public resources towards the development of the (forest-based) bioeconomy that is sufficient in quantity and stable over time | 3.5 | 3.4 | 2.7 |
| | Legislation that does not hinder the development of the (forest-based) bioeconomy in the region | 3.8 | 2.8 | 2.6 |
| | Public support for products and solutions generated from (forest-based) bioeconomy initiatives | 3.2 | 2.7 | 2.4 |
| **R&D&i** | Public and public-private investment on R&D&i that can be directed towards the development of the (forest-based) bioeconomy | 4.4 | 3.0 | 2.9 |
| **Training and talent** | Existence of training and talent programmes in the region that can be used to boost the development of the (forest-based) bioeconomy | 3.6 | 3.1 | 2.7 |
| | Education and awareness-raising campaigns and actions in the region that can be used to boost the development of the (forest-based) bioeconomy | 3.8 | 2.4 | 2.7 |
| **Ecosystem for entrepreneurship** | An ecosystem favourable for entrepreneurship and intrapreneurship in the region | 3.4 | 4.2 | 4.0 |
| **Green public procurement** | Existence of green public procurement in the region that can be used to boost the development of the (forest-based) bioeconomy | 3.2 | 2.7 | 2.6 |
| **Regional networks** | The region participates in European and international regional networks that can be used to boost the development of the (forest-based) bioeconomy | 3.4 | 3.8 | 3.8 |
| | *Supply* | | | |
| **Entrepreneurial capacities** | Existence of capital (investment funds, financial institutions) that can support (forest-based) bioeconomy initiatives. The region displays capabilities that efficiently explore, integrate, and exploit untapped business opportunities, such as those provided by the (forest-based) bioeconomy | 3.0 | 2.7 | 2.7 |
| **Existence of clusters** | Existence of clusters related to the (forest-based) bioeconomy | 4.0 | 4.0 | 2.0 |
| | *Demand* | | | |
| **Market awareness and demand** | The society understands the concept of the (forest-based) bioeconomy and supports its implementation in the region. The market supports and demands products and services generated from (forest-based) bioeconomy initiatives | 3.1 | 2.7 | 2.6 |
| | *Biomass* | | | |
| **Biomass** | Sufficient and constant supply of (forest) biomass in the region in term of quantity and rate of generation | 4.6 | 3.1 | 3.7 |
| | Sufficient and constant supply of (forest) biomass in the region in terms of quality and price | 4.5 | 3.1 | 3.3 |
| **MEAN SCORE** | | **3.7** | **3.1** | **3.0** |

As shown in Table 3, in our study, North Karelia and the Basque Country obtained the highest (3.7 points) and lowest (3.0 points) score, respectively. Regarding the qualitative information extracted from the interviews, no explicit negative comments on any of the drivers (deployed in 15 items) were expressed by the experts from North Karelia. In actual fact, all of them agreed on the solid development of forest-based bioeconomy in the region. On the other hand, the experts from North Rhine-Westphalia indicated the

following obstacles for forest-based bioeconomy development in the region: (i) the existing legislation does not strongly favour the development of the forest-based bioeconomy; (ii) insufficient education and awareness-raising campaigns and actions aimed at the forest-based bioeconomy development; and (iii) absence of the necessary capital and support to boost the development of the forest-based bioeconomy in the region. In relation to the Basque Country, a variety of barriers for forest-based bioeconomy development were explicitly mentioned by the experts: (i) the allocation of public resources to stimulate such development is still insufficient; (ii) the current legislation does not strongly favour the development of the forest-based bioeconomy; (iii) the absence of ad hoc training and talent programmes on this matter; (iv) the market and the consumers are not aware of the potential advantages associated to the use of bio-based products; (v) the absence of the necessary private capital to boost the development of the forest-based bioeconomy; and (vi) a lack of a supportive societal environment for the development of the forest-based bioeconomy, since an important part of the Basque society is more interested in other forest ecosystem services (e.g., recreation, carbon sequestration, biodiversity conservation). Certainly, (forest) ecosystems provide an assortment of services that are essential to human survival, well-being, and health [35,36]. Interestingly, the concept of ecosystem services can help bridge the scientific–economic-policy-making divide [37].

3.2.2. Secondary Sources of Information: Literature Review

This primary information provided by experts was confronted with secondary information extracted from our literature review. The main sources of secondary information consulted here are shown in Table A2. Table 4 shows the degree of development of forest-based bioeconomy according to the secondary sources of information.

**Table 4.** Scores (from 1 to 3) for the drivers extracted from secondary sources of information.

| Driver | Item | North Karelia | North Rhine-Westphalia | Basque Country |
|---|---|---|---|---|
| *Institutional* | | | | |
| **Government plans and policies** | In North Karelia and the Basque Country, government plans and policies directed towards the development of the (forest-based) bioeconomy have been in place for a longer time | 3.0 | 2.0 | 3.0 |
| **R&D&i** | North Karelia and North Rhine-Westphalia have included the forest-based bioeconomy in their RIS3 strategies. In the Basque Country, capacities for R&D&i are very high, but they are insufficiently applied to bioeconomy development. The Basque Country has not included the (forest-based) bioeconomy in its RIS3 strategy | 3.0 | 3.0 | 2.0 |
| **Training and talent** | In North Karelia, they have implemented training plans for the development of the forest-based bioeconomy. In North Rhine-Westphalia, they are currently working on it. In the Basque Country, there are no ad hoc training programmes on the (forest-based) bioeconomy | 3.0 | 2.0 | 1.0 |
| **Ecosystem for entrepreneurship** | North Karelia has an accelerator–incubator focused on the development of the (forest-based) bioeconomy. North Rhine-Westphalia has similar infrastructures. In the Basque Country, there are no ad hoc (forest-based) bioeconomy entrepreneurship programmes | 3.0 | 3.0 | 1.0 |

**Table 4.** *Cont.*

| Driver | Item | North Karelia | North Rhine-Westphalia | Basque Country |
|---|---|---|---|---|
| *Institutional* | | | | |
| **Green public procurement** | The three regions employ green public procurement (North Karelia and the Basque Country appear to be somewhat more advanced in this matter than North Rhine-Westphalia) | 2.0 | 1.0 | 2.0 |
| **Regional networks** | The three regions have very active and well-positioned regional networks | 3.0 | 3.0 | 3.0 |
| *Supply* | | | | |
| **Entrepreneurial capacities** | The three regions have strong and diversified entrepreneurial capacities | 3.0 | 3.0 | 3.0 |
| **Existence of clusters** | North Karelia and North Rhine-Westphalia have clusters specialized on the (forest-based) bioeconomy. In the Basque Country, the number of clusters is very high, but the degree of focus on bioeconomy is still insufficient | 3.0 | 3.0 | 2.0 |
| *Demand* | | | | |
| **Market awareness and demand** | No data on this matter have been found in the literature | - | - | - |
| *Biomass* | | | | |
| **Biomass** | North Karelia has a much higher forest biomass than North Rhine-Westphalia. However, both are in a good situation in terms of biomass quantity and replacement rate. The Basque Country offers a good supply of forest biomass in terms of quantity and quality | 3.0 | 2.0 | 3.0 |
| **MEAN SCORE** | | **2.9** | **2.4** | **2.2** |

Again, as shown in Table 4, North Karelia and the Basque Country obtained the highest (2.9 points) and lowest (2.2 points) scores, respectively, according to the secondary information extracted from the literature review.

When adding the scores provided by the primary and secondary sources of information, it was concluded that North Karelia has a high development of forest-based bioeconomy (3.7 + 2.9 = 6.6 points), while North Rhine-Westphalia (3.1 + 2.4 = 5.5 points) and the Basque Country (3.0 + 2.2 = 5.2 points) have a medium level of development. This conclusion coincides with the information given to us by the Bioregions Facility of the European Forest Institute (personal communication), proving the validity of our assessment.

## 4. Conclusions

In this study, we have assessed the development of the forest-based bioeconomy in three European region, by appraising the status of 10 identified key drivers through primary (interviews with experts) and secondary (literature review) sources of information, finding out that North Karelia and the Basque Country obtained the highest and lowest scores, respectively. This study has been carried out in only three European regions, so, in consequence, it certainly cannot be considered representative of, nor directly extrapolatable to, other European or non-European regions. Nonetheless, the three studied regions belong to the Bioregions Facility of the European Forest Institute, an initiative designed to support innovation, networking, and policy learning related to the development of the forest-based bioeconomy, which develops joint strategies and actions, promotes forest-based bioeconomy business and innovation, and hosts capacity-building activities and

knowledge exchanges. In turn, the European Forest Institute, an international organization with around 130 associate and affiliate member organisations in 40 countries, facilitates forest-related networking and promotes the dissemination of relevant information on forests and forestry. Therefore, via the Bioregions Facility of the European Forest Institute, the methodology used and the results obtained in this study will have a broader audience and resonance. On the other hand, the 10 key drivers could have been weighed in the order of their relative importance according to previously defined criteria, but we reckon that such weighing might also have some undesirable consequences, especially when dealing with relatively new transitions (e.g., the bioeconomy transition), such as the lack of recognition of the imperative need to initially cover all the different aspects encompassed by those 10 drivers to successfully accomplish the targeted long-term transition towards a forest-based bioeconomy. As a matter of fact, our initial proposition was that, for the forest-based bioeconomy to successfully and sustainably develop in the long-term in a European region, many different drivers must be present and/or stimulated in unison by a variety of stakeholders (regional governments, research institutions, companies, consumers, etc.). Since it cannot be predicted a priori which specific drivers will be more effective to boost the targeted transition, we have proposed here to initially pay the same attention (i.e., to assign an equal value) to all 10 drivers, without closing the door to the possibility of weighting them differently once the effectiveness of each of them (in relation to the development of the the forest-based bioeconomy in the region under study) has been verified. Importantly, in this sense, it is plausible that such weighting would be different depending on the specific casuistry of each region. In any case, in order to ensure the optimum development of the forest-based bioeconomy in a given European region, it is critical to always emphasize the unnegotiable significance of the sustainability of the biomass resource base, the sustainability of production processes, the sustainability of consumption patterns, and the circularity of material fluxes. Certainly, for the bioeconomy to succeed, it must be categorically and unconditionally based on firm sustainability principles and must pave the way towards a more equitable, inclusive, and prosperous society, reconciling economic growth with environmental conservation.

**Author Contributions:** Conceptualization, A.A. and C.G.; investigation, L.B. and O.U.; data curation: L.B., O.U., S.B., T.O. and B.S.; writing—original draft preparation, C.G., O.U. and N.G.; writing—review and editing, C.G., O.U., N.G., L.B., S.B., T.O., B.S., I.M.d.A. and A.A. All authors have read and agreed to the published version of the manuscript.

**Funding:** This research received no external funding.

**Data Availability Statement:** Not applicable.

**Conflicts of Interest:** The authors declare no conflict of interest.

## Appendix A

**Table A1.** List of publications.

| Year | Authors | Title | Ref. |
|------|---------|-------|------|
| 2020 | Falcone, P.M.; Tani, A.; Tartiu, V.E.; Imbriani, C. | Towards a sustainable forest-based bioeconomy in Italy: Findings from a SWOT analysis | [5] |
| 2020 | Robert, N.; Jonsson, R.; Chudy, R.; Camia, A. | The EU bioeconomy: Supporting an employment shift downstream in the wood-based value chains? | [21] |
| 2020 | D'Amato, D.; Veijonaho, S.; Toppinen, A. | Towards sustainability? Forest-based circular bioeconomy business models in Finnish SMEs | [22] |

| Year | Authors | Title | Ref. |
|------|---------|-------|------|
| 2020 | Poduška, Z.; Nedeljković, J.; Nonić, D.; Ratknić, T.; Ratknić, M.; Zivojinović, I. | Intrapreneurial climate as momentum for fostering employee innovativeness in public forest enterprises | [23] |
| 2020 | Kuckertz, A.; Berger, E.; Brändle, L. | Entrepreneurship and the sustainable bioeconomy transformation | [24] |
| 2020 | Fradj, N.B.; Jayet, P.A.; Rozakis, S.; Georgenta, E.; Jedrejek, A. | Contribution of agricultural systems to the bioeconomy in Poland: Integration of willow in the context of a stylised CAP diversification | [25] |
| 2020 | Lawrence, A.; Wong, J.L.G; Molteno, S. | Fostering social enterprise in woodlands: Challenges for partnerships supporting social innovation | [26] |
| 2020 | Baycheva-Merger, T.; Sotirov, M. | The politics of an EU forest information system: Unpacking distributive conflicts associated with the use of forest information | [27] |
| 2020 | Brunnhofer, M.; Gabriella, N.; Shöggl.; Stern, T.; Posch, A. | The biorefinery transition in the European pulp and paper industry—A three-phase Delphi study including a SWOT-AHP analysis | [28] |
| 2020 | Tittor, A. | The changing drivers of oil palm cultivation and the persistent narrative of 'already degraded land'. Insights from Nicaragua | [29] |
| 2020 | Lazarevic, D.; Kautto, P.; Antikainen, R. | Finland's wood-frame multi-storey construction innovation system: Analysing motors of creative destruction | [30] |
| 2020 | Padró, R.; Tello, E.; Marco, I.; Olarieta, J.R.; Grasa, M.M.; Font, C. | Modelling the scaling up of sustainable farming into Agroecology Territories: Potentials and bottlenecks at the landscape level in a Mediterranean case study | [31] |
| 2019 | Devaney, L.; Lles, A. | Scales of progress, power and potential in the US bioeconomy | [32] |
| 2019 | Hurmekoski, E.; Lovrić, M.; Lovrić, N.; Hetemäki, L.; Winkel, G. | Frontiers of the forest-based bioeconomy—A European Delphi study | [33] |
| 2019 | Bonsu, N.O.; McMahon, B.J.; Meijer, S.; Young, J.C.; Keane, A.; Dhubháin, A.N. | Conservation conflict: Managing forestry versus hen harrier species under Europe's Birds Directive | [34] |
| 2019 | Colombo, L.A.; Pansera, M.; Owen, R. | The discourse of eco-innovation in the European Union: An analysis of the Eco-Innovation Action Plan and Horizon 2020 | [35] |
| 2019 | Hernik, J.; Noszczyk, T.; Rutkowska, A. | Towards a better understanding of the variables that influence renewable energy sources in eastern Poland | [36] |
| 2019 | Bauer, F.; Fuenfschilling, L. | Local initiatives and global regimes—Multi-scalar transition dynamics in the chemical industry | [37] |
| 2018 | Koukios, E.; Monteleone, M.; Texeira, M.J.; Charalambous, A.; Girio, F.; López Hernández, E.; Mannelli, S.; Parajó, J.C.; Polycarpuo, P.; Zabaniotou, A. | Targeting sustainable bioeconomy: A new development strategy for Southern European countries. The Manifesto of the European Mezzogiorno | [38] |
| 2018 | Bauer, F. | Narratives of biorefinery innovation for the bioeconomy: Conflict, consensus or confusion? | [39] |

**Table A1.** *Cont.*

| Year | Authors | Title | Ref. |
|------|---------|-------|------|
| 2018 | Purkus, A.; Hagemann, N.; Bedtke, N.; Gawel, E. | Towards a sustainable innovation system for the German wood-based bioeconomy: Implications for policy design | [40] |
| 2018 | Zabaniotou, A. | Redesigning a bioenergy sector in EU in the transition to circular waste-based Bioeconomy—A multidisciplinary review | [41] |
| 2018 | Kokkonen, K.; Ojanen, V. | From opportunities to action—An integrated model of small actors' engagement in bioenergy business | [42] |
| 2018 | Hurmekoski, E.; Pykäläinen, J.; Hetemäki, L. | Long-term targets for green building: Explorative Delphi backcasting study on wood-frame multi-story construction in Finland | [43] |
| 2018 | Ingrao, C.; Bacenetti, J.; Bezama, A.; Blok, V.; Goglio, P.; Koukios, E.; Lindner, M.; Nemecek, T.; Siracusa, V.; Zabaniotou, A.; Huisingh, D. | The potential roles of bio-economy in the transition to equitable, sustainable, post fossil-carbon societies: Findings from this virtual special issue | [44] |
| 2017 | Giurca, A.; Späth, P. | A forest-based bioeconomy for Germany? Strengths, weaknesses and policy options for lignocellulosic biorefineries | [45] |
| 2017 | D'Amato.; Droste, N.; Allen.; Kettunen, M.; Lähtinen, K.; Korhonen, J.; Leskinen, P.; Matthies, B.D.; Toppinen, A. | Green, circular bio economy: A comparative analyses of sustainability avenues. | [46] |
| 2017 | Giurca, A.; Metz, T. | A social network analysis of Germany's wood-based bioeconomy: Social capital and shared beliefs | [47] |
| 2017 | Mossberg, J.; Söderholm, P.; Hellsmark, H.; Nordqvist, S. | Crossing the biorefinery valley of death? Actor roles and networks in overcoming barriers to a sustainability transition | [48] |
| 2017 | Živojinović, I.; Nedeljković, J.; Stojanovski, V.; Japelj, A.; Nonić, D.; Weiss, G. | Non-timber forest products in transition economies: Innovation cases in selected SEE countries | [49] |
| 2016 | Hagemann, N.; Gawel, E.; Purkus, A.; Pannicke, N.; Hauck, J. | Possible futures towards a wood-based bioeconomy: A scenario analysis for Germany | [50] |
| 2016 | Pätäri, S.; Tuppura, A.; Toppinen, A.; Korhonen, J. | Global sustainability megaforces in shaping the future of the European pulp and paper industry towards a bioeconomy | [51] |
| 2016 | Leban, V.; Malovrh, S.P.; Stirn, L.Z.; Krč, J. | Forest biomass for energy in multi-functional forest management: Insight into the perceptions of forest-related professionals | [52] |
| 2016 | Hellsmark, H.; Mossberg, J.; Söderholm, P.; Frishammar, J. | Innovation system strengths and weaknesses in progressing sustainable technology: the case of Swedish biorefinery development | [53] |
| 2014 | Haatanen, A.; den Herder, M.; Leskinen, P.; Lindner, M.; Kurttila, M.; Salminen, O. | Stakeholder engagement in scenario development process—Bioenergy production and biodiversity conservation in eastern Finland | [54] |
| 2014 | Hurmekoski, E.; Hetemäki, L. | Studying the future of the forest sector: Review and implications for long-term outlook studies | [55] |
| 2014 | Näyhä, A. | Strategic change in the forest industry toward the biorefining business | [56] |

**Table A1.** *Cont.*

| Year | Authors | Title | Ref. |
|------|---------|-------|------|
| 2014 | Kleinschmit, D.; Lindstad, B.H.; Thorsen, B.J.; Toppinen, A.; Roos, A; Baardsen, S.. | Shades of green: A social scientific view on bioeconomy in the forest sector | [57] |
| 2013 | Staffas, L.; Gustavsson, M.; McCormick, K. | Strategies and policies for the bioeconomy and bio-based economy: An analysis of official rational approaches | [58] |

**Table A2.** Secondary sources of information consulted to assess the development of forest-based bioeconomy in the three regions.

| Region | Secondary Sources of Information * |
|--------|-----------------------------------|
| North Karelia | <ul><li>https://www.stat.fi/tup/suoluk/suoluk_kansantalous_en.html—Gross domestic product per capita</li><li>Smart Specialisation Report in North Karelia por Regional Council of North Karelia, 2014</li><li>Global Education Park Finland Report, https://www.globaleducationparkfinland.fi.</li><li>Forest Bioeconomy in North Karelia https://www.pohjois-karjala.fi/documents/78299/3959112/Forest%20Bioeconomy%20in%20North%20Karelia%202019.pdf/1c7d4af2-3d96-9bd3-70f5-06c669688f47</li><li>North Karelia & Smart forest bioeconomy: https://www.pohjois-karjala.fi/documents/78299/3959112/Smart%20Forest%20Bioeconomy%20Brochure.pdf/a4f741eb-8b14-0845-362d-e58721cbb695</li><li>https://www.pohjois-karjala.fi/en/web/smarteast.fi/home</li><li>POKAT 2021. North KareliA's Regional Strategic Programme for 2018-2021: https://www.pohjois-karjala.fi/documents/33565/34607/POKAT+2021+Summary.pdf/80583d66-9e7b-d4f6-8e90-3dc0a4552dac?version=1.0</li><li>Climate and Energy Programme of North Karelia 2020: A summary. https://www.pohjois-karjala.fi/documents/78299/173745/Climate%20and%20Energy%20Programme%20of%20North%20Karelia.pdf/a327551e-2606-497f-9db9-afacb99ee5e3</li></ul> |
| North Rhine-Westphalia | <ul><li>North Rhine-Westfalia Innovation Strategy in scope of EU-structure fund2014-2020. https://s3platform.jrc.ec.europa.eu/documents/20182/229963/DE_NRW_RIS3_Final.pdf/4ad1bd7d-5610-41b9-bf7d-e6eb10c709ed</li><li>https://www.wald-und-holz.nrw.de/ueber-uns/en/about-us</li><li>https://proholz.nrw/wp-content/uploads/2020/12/Walddaten.pdf</li><li>https://ec.europa.eu/growth/tools-databases/regional-innovation-monitor/base-profile/north-rhine-Renania del Norte-Westfalia</li><li>https://www.researchgate.net/publication/326248291_Technological_relatedness_knowledge_space_and_smart_specialisation_The_case_of_German</li></ul> |
| Basque Country | <ul><li>Basque Agency for Innovation. https://www.innobasque.eus/</li><li>Basque Strategy for Circular Economy. https://www.euskadi.eus/documentacion/2020/estrategia-de-economia-circular-de-euskadi-2030/web01-a2ingkut/es/</li><li>Basque Wood Association. https://baskegur.eus/forestal-madera/</li><li>Basque Country Clusters. https://www.euskadi.eus/gobierno-vasco//contenidos/noticia/clusters_sectoriales_09/es_clusters/clusters.html</li><li>The Basque Business Development Agency. https://www.spri.eus/en/invest-in-the-basque-country/</li><li>Basque Foundation for Science. https://www.ikerbasque.net/</li><li>Basque Technology Parks. https://parke.eus/en/</li><li>University of the Basque Country. https://www.ehu.eus/en/en-home</li><li>University of Deusto. https://www.deusto.es/cs/Satellite/deusto/en/university-deusto</li><li>Mondragon University. https://www.mondragon.edu/en/home</li><li>NEIKER-Basque Institute of Agricultural Research and Development. https://neiker.eus/en/</li><li>Basque Research and Technology Alliance. https://www.brta.eus/en/home</li><li>HAZI Foundation-Rural, coastal and food development. https://www.hazi.eus/es/</li><li>Basoa Fundazioa-Confederation of Foresters of the Basque Country. http://basoa.org/es/</li><li>ACLIMA-Basque Environment Cluster. https://aclima.eus/en/</li><li>IHOBE-Public Society of Environmental Management of the Basque Government. https://www.ihobe.eus/about-ihobe</li></ul> |

\* Accessed on 1 April 2021.

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
