# Peer review of "Assessment of the Development of Forest-Based Bioeconomy in European Regions"

_sustainability, doi:10.3390/su14084747_

Round 1

Reviewer 1 Report

Re: Comments: Assessment of the development of forest-based bioeconomy in European regions

The manuscript makes an interesting point by identifying and analysing the factors relevant for nurturing a sustainable forest-based bioeconomy across three European regions with three different forest research endowments. The comparative analysis is illuminating and should be published, pending addressing three essential issues: writing a theoretical perspective, improving the methodology section, and discussing the broader implications of the findings for building competitive and sustainable forest-based within the EU. Further information on each point is provided below.

Theoretical framing. The manuscript currently lacks an explicit description of the authors’ theoretical perspective. For the kind of qualitative analysis the authors note in their introduction, I think it would be essential to have a theoretical aspect. A good starting point might be to consider the socio-technical transitions embodied in the Multi-Level Perspective (MLP) framework. The MLP, with its engagement of evolutionary economics, neo-institutional economics, sociology of innovation and more recently, political and discourse theory, could give the authors a broad lens to explain and draw resonance from the ten factors they identified as drivers of the European forest-based bioeconomy.

Methods: The authors argue that they employed a qualitative analysis both in the abstract and introduction; for example, ‘in this study, using the qualitative case method, we evaluated the development of forest-based bioeconomy in three European regions’ (see lines 87 & 88). Yet, in the methodology section, the authors appear to use a quantitative approach in assessing the extent to which forest-based bioeconomy had developed in the case study communities. Could the authors clarify what informed their decision or update the earlier portion of the paper to reflect the methods they used?

Another point on the methods is the lack of clarity on how the authors conducted their literature search. If the authors are convinced that their identified categories are exhaustive, then there should be some justification for the material reviewed and how they were reviewed. The authors write that ‘we carefully sifted through the material to discard information that was considered to be too general, vague, controversial or indeterminate’ (lines 125-126). However, there is no information on how the sifting was done or the type of indicators used to cluster the multiple drivers. Again, this point highlights why the authors need to include a theoretical aspect to the paper. Such a section will provide a broader context to guide their analysis, improving readers’ appreciation of the authors’ contribution to knowledge.

Another point on the methods relates to the authors interviewed: researchers, decision-makers, and businessmen. It is good practice to provide a table of the various actors interviewed and their relative numbers in qualitative research. Besides, was there saturation in the data gathered? At what point did the authors terminate the interviews? Moreover, the authors did not clarify the type of interviews conducted, whether they recorded them, and more importantly, how they analysed information from the interview, including the number of people that did the analyses. Such information guarantees scientific rigour in qualitative research and should be provided.

Finally, given biomass availability is a core of the drivers, could the authors elaborate if they interviewed any forests owners to have their perspective and rating on the issues. In my opinion, they are essential actors whose perspectives matter on the topic under consideration. If they are part of the businessmen, could the authors make this more explicit? Thus, the authors should aim to resolve this.

This might be a useful reference for the authors in clarifying the methods: Malterud, K. (2001). Qualitative research: standards, challenges, and guidelines. The Lancet, 358(9280), 483–488. doi:10.1016/s0140-6736(01)05627-6

Results, discussion, and conclusion

The authors present their results and discuss them well. However, I miss the broader resonance of the work for the bioeconomy transition. I think it is not adequate only to identify and compare the driver of the bioeconomy from different regions and actor perspectives. What do the findings mean for EU Bioeconomy policy, related investments at the territorial, EU and international level or actors from whom the EU imports biomass? Providing such information will make the final state of the conclusion worthwhile:

“Certainly, for bioeconomy to succeed, it must be based on firm sustainability principles and must pave the way towards a more equitable, inclusive and prosperous society, reconciling economic growth with environmental conservation (lines 346-347).

Author Response

We sincerely thank the Reviewers for their valuable and constructive comments, which have been very helpful in improving the manuscript. The manuscript has been modified following all their comments.

REVIEWER #1: We sincerely thank Reviewer #1 for carefully reviewing our manuscript and for her/his thoughtful feedback.

COMMENT: The manuscript currently lacks an explicit description of the authors’ theoretical perspective. For the kind of qualitative analysis the authors note in their introduction, I think it would be essential to have a theoretical aspect. A good starting point might be to consider the socio-technical transitions embodied in the Multi-Level Perspective (MLP) framework. The MLP, with its engagement of evolutionary economics, neo-institutional economics, sociology of innovation and more recently, political and discourse theory, could give the authors a broad lens to explain and draw resonance from the ten factors they identified as drivers of the European forest-based bioeconomy.

ANSWER: Following the Reviewer´s comment, we have now included the theoretical perspective. Please see lines 65-111. We sincerely thank the Reviewer for her/his suggestion regarding the MLP framework, which has been very useful for developing the theoretical perspective (thank you very much for pointing out this possibility; to be frank, we did not know the MLP framework and it has been a most interesting “finding” for our group).

COMMENT: The authors argue that they employed a qualitative analysis both in the abstract and introduction; for example, ‘in this study, using the qualitative case method, we evaluated the development of forest-based bioeconomy in three European regions’ (see lines 87 & 88). Yet, in the methodology section, the authors appear to use a quantitative approach in assessing the extent to which forest-based bioeconomy had developed in the case study communities. Could the authors clarify what informed their decision or update the earlier portion of the paper to reflect the methods they used?

ANSWER: The Reviewer is right. We used both: qualitative analysis for the definition of the ten factors (via secondary sources of information) and quantitative analysis when asking the interviewees (primary sources of information) to score the abovementioned factors (as well as when we scored them). This fact has now been corrected in the text.

COMMENT: Another point on the methods is the lack of clarity on how the authors conducted their literature search. If the authors are convinced that their identified categories are exhaustive, then there should be some justification for the material reviewed and how they were reviewed. The authors write that ‘we carefully sifted through the material to discard information that was considered to be too general, vague, controversial or indeterminate’ (lines 125-126). However, there is no information on how the sifting was done or the type of indicators used to cluster the multiple drivers. Again, this point highlights why the authors need to include a theoretical aspect to the paper. Such a section will provide a broader context to guide their analysis, improving readers’ appreciation of the authors’ contribution to knowledge.

ANSWER: Following the Reviewer´s comment, additional information has been included in the text to clarify the performed literature review. Please see lines 177-192.

COMMENT: Another point on the methods relates to the authors interviewed: researchers, decision-makers, and businessmen. It is good practice to provide a table of the various actors interviewed and their relative numbers in qualitative research. Besides, was there saturation in the data gathered? At what point did the authors terminate the interviews? Moreover, the authors did not clarify the type of interviews conducted, whether they recorded them, and more importantly, how they analysed information from the interview, including the number of people that did the analyses. Such information guarantees scientific rigour in qualitative research and should be provided.

ANSWER: Following the Reviewer´s comment, additional information has been included in the text to clarify the performed interviews. Please see lines 213-217 and 249-261, as well as Table 1.

COMMENT: Finally, given biomass availability is a core of the drivers, could the authors elaborate if they interviewed any forests owners to have their perspective and rating on the issues. In my opinion, they are essential actors whose perspectives matter on the topic under consideration. If they are part of the businessmen, could the authors make this more explicit? Thus, the authors should aim to resolve this.

ANSWER: Unfortunately, we did not interview any forest owners. Following the Reviewer´s comment, we have now stated in the text the importance of taking into consideration the forest owners´ perspective when studying the development of forest-based bioeconomy in a given region (please see lines 256-260). Besides, data on biomass availability in the three European regions have been included (please see lines 167-174).

COMMENT: This might be a useful reference for the authors in clarifying the methods: Malterud, K. (2001). Qualitative research: standards, challenges, and guidelines. The Lancet, 358(9280), 483–488. doi:10.1016/s0140-6736(01)05627-6

ANSWER: Thank very much for suggesting this interesting reference. It has been carefully studied and discussed in the text (please see lines 228-243).

COMMENT: The authors present their results and discuss them well. However, I miss the broader resonance of the work for the bioeconomy transition. I think it is not adequate only to identify and compare the driver of the bioeconomy from different regions and actor perspectives. What do the findings mean for EU Bioeconomy policy, related investments at the territorial, EU and international level or actors from whom the EU imports biomass? Providing such information will make the final state of the conclusion worthwhile: “Certainly, for bioeconomy to succeed, it must be based on firm sustainability principles and must pave the way towards a more equitable, inclusive and prosperous society, reconciling economic growth with environmental conservation (lines 346-347).

ANSWER: Following the Reviewer´s comment, additional information has been included in the text to emphasize the broader resonance of our findings (please see lines 454-466).

Reviewer 2 Report

This is an interesting and up-to-date paper, which adds to the debate on the implementation of a bioeconomy. The methods and results were well explained.

Author Response

We sincerely thank the Reviewers for their valuable and constructive comments, which have been very helpful in improving the manuscript. The manuscript has been modified following all their comments.

REVIEWER #2: We sincerely thank Reviewer #2 for carefully reviewing our manuscript and for her/his thoughtful feedback.

COMMENT: This is an interesting and up-to-date paper, which adds to the debate on the implementation of a bioeconomy. The methods and results were well explained.

ANSWER: Thank you very much for your kind comments.

Reviewer 3 Report

The idea of the article is interesting, however the presentation and results of what authors have one is insufficient. See below:

1) terminology used in the paper is not united. I do not think it is possible to mix up terms bioeconomy, forest-based bioeconomy, sustainable circular bioeconomy etc. without further specifying their content; if the whole article deals with F-B bioeconomy, then it should be also defined.

2) Material and methods

- are very hard to follow

ii) what does "strong forestry sector" mean?

l. 154 - I do not think it is a good idea to cite reference from 1975 regarding the procedure of who should be the interviewees.

l. 158... - there is no really exact information on who were the people who you interviewed. How many people you interviewed? There is no information on how these people were linked between - knowledge (l. 158 to 162) and work positions (§.162-166). I mean being a businessman does not mean I have knowledge of forestry and even not on bioeconomy. This might be your own choice, that is fine, but still some more info is needed - at least on work positions - should be provided dividing the people according to the regions (e.g. Ms. XY - state forestry company named XY - chief director - North Karelia).

I do not get why there are different scales (1-5) and 1-3. It might be, when expert opinion are more important but - it must be explained.

3) Results

I think it is hard to give a proper scoring on how some documents/policies, training programmes etc. are connected to bioeconomy. That means that multi-stakeholder approach (more people who gave the opinion) were incorporated in the survey - or - the reader need to know that the expert can cover all these fields with own individual opinion. (That comes back to my question about the selection of people, who were they etc.).

Sorry, but scoring according to normal arithmetic mean when there are different scales, different number of individual points in items (e.g. institutional - 6, supply - 2) is totally insufficient to me. Did you think about some of them that might be much more important than the other factors?

4) Conclusions

As a matter of papers, the lenght of the conclusion is short. I do not think it is a right way to present that one region is "the best" and other one "the worst" when you are only evaluating 3 regions.

Overall, it seems to me as a mixture of literature review with case study applications with nice potential but badly described methodology, usage of ununited terminology and very hard-to-follow structure of presented information.

Author Response

We sincerely thank the Reviewers for their valuable and constructive comments, which have been very helpful in improving the manuscript. The manuscript has been modified following all their comments.

REVIEWER #3: We sincerely thank Reviewer #3 for carefully reviewing our manuscript and for her/his thoughtful feedback.

COMMENT: 1) terminology used in the paper is not united. I do not think it is possible to mix up terms bioeconomy, forest-based bioeconomy, sustainable circular bioeconomy etc. without further specifying their content; if the whole article deals with F-B bioeconomy, then it should be also defined.

ANSWER: The terminology has been clarified throughout the text as suggested by the Reviewer.

COMMENT: What does "strong forestry sector" mean?

ANSWER: That expression has been clarified. Please see lines 156-158.

COMMENT: l. 154 - I do not think it is a good idea to cite reference from 1975 regarding the procedure of who should be the interviewees.

ANSWER: More recent references have been included. Please see lines 222-243.

COMMENT: l. 158... - there is no really exact information on who were the people who you interviewed. How many people you interviewed? There is no information on how these people were linked between - knowledge (l. 158 to 162) and work positions (§.162-166). I mean being a businessman does not mean I have knowledge of forestry and even not on bioeconomy. This might be your own choice, that is fine, but still some more info is needed - at least on work positions - should be provided dividing the people according to the regions (e.g. Ms. XY - state forestry company named XY - chief director - North Karelia).

ANSWER: Following the Reviewer´s comment, additional information has been included in the text to clarify the performed interviews. Please see lines 213-217 and 249-261, as well as Table 1.

COMMENT: I do not get why there are different scales (1-5) and 1-3. It might be, when expert opinion are more important but - it must be explained.

ANSWER: The choice of the different scales has been clarified. Please see lines 267-281.

COMMENT:  Results. I think it is hard to give a proper scoring on how some documents/policies, training programmes etc. are connected to bioeconomy. That means that multi-stakeholder approach (more people who gave the opinion) were incorporated in the survey - or - the reader need to know that the expert can cover all these fields with own individual opinion. (That comes back to my question about the selection of people, who were they etc.). Sorry, but scoring according to normal arithmetic mean when there are different scales, different number of individual points in items (e.g. institutional - 6, supply - 2) is totally insufficient to me. Did you think about some of them that might be much more important than the other factors?

ANSWER: We did consider weighing the different factors but, finally, we opted for the arithmetic mean for the reasons now stated in lines 466-482.

COMMENT: Conclusions. As a matter of papers, the length of the conclusion is short. I do not think it is a right way to present that one region is "the best" and other one "the worst" when you are only evaluating 3 regions. Overall, it seems to me as a mixture of literature review with case study applications with nice potential but badly described methodology, usage of ununited terminology and very hard-to-follow structure of presented information.

ANSWER: The Conclusion section has been modified following the Reviewer´s comment. In addition, the manuscript has been completely revised in an attempt to resolve the limitations mentioned by the Reviewer. Please see lines 449-489.

Round 2

Reviewer 3 Report

Although I still lack some interconnections in the paper as such, I do appreciate author´s efforts to improve the article according to my suggestions and I recommend it for being published.